# Early ethics: Exploring moral intuition and maternal Influence in preschool children

**María G. Jean-Tron[1]<sup></sup>, Juan Garduño-Espinosa[1]‡\*, Diana Ávila-Montiel[2]<sup></sup>,
Gina C. Chapa-Koloffon[1]<sup></sup>, Oscar A. Resendez-Berber[2]<sup></sup>, Elizabeth Cruz Cruz[2]<sup></sup>,
Guillermo Salinas-Escudero[3]<sup></sup>, Onofre Muñoz-Hernández[2]‡**

**1** Evidence-Based Medicine and Applied Ethics Research Unit, Hospital Infantil de México Federico Gómez, Mexico City, Mexico, **2** Research Department, Hospital Infantil de México Federico Gómez, Mexico City, Mexico, **3** Center for Economic and Social Studies in Health, Hospital Infantil de México Federico Gómez, Mexico City, Mexico

These authors contributed equally to this work
‡ These authors also contributed equally to this work
\* juan.gardunoe@gmail.com

## Abstract

The purpose of this study was to evaluate the moral intuitions of mothers and children aged 3–6 (n = 75). Methods: Five dilemmas were applied. The agreement between mother and child responses was evaluated, as were trends in agreement between girls and boys. Kappa statistics and Spearman's correlation analyses were conducted. McNemar's test was administered to assess the Double Effect Doctrine and the Contact Principle. Outcome: In general, children responded to moral dilemmas similarly to their mothers. However, no significant agreement was found between mothers and children when evaluating each dilemma. Although the children's answer patterns were similar to those of their mothers, the presence of neither the Double Effect Doctrine nor the Contact Principle could be identified in children. Conclusions: While there are some similarities between preschoolers and their mothers when responding to moral dilemmas, the integration of deontological principles in the resolution of ethical dilemmas in the children studied has not been achieved. Mothers in the study use these principles, which support the previous related evidence.

## Introduction

### Universal moral grammar and deontological principles

The Universal Moral Grammar (UMG) posits that moral judgments are based on innate, operative principles that precede culture and language and automatically guide our moral intuitions [1,2].

There are three main deontological principles included within UMG. The first, known as the Action Principle or Omission Bias, suggests that the damage caused by an action has a worse moral effect than the damage caused by a lack of interference,

**Data availability statement:** All relevant data are within the paper and its Supporting Information files.

**Funding:** The author(s) received no specific funding for this work.

**Competing interests:** The authors have declared that no competing interests exist.

i.e., by omission. The Contact Principle suggests that making physical contact with someone else has a more negative effect than causing the same damage without physical contact. The third principle, the Doctrine of Double Effect or the Intention Principle, states that causing purposeful damage to achieve a goal is morally worse than causing harm as a secondary effect of achieving that goal [3,4].

According to the GMU, these moral principles evolved to strengthen cooperation and regulate conflicts between humans; their automatic internalization promotes social cohesion and sets the universal ethical foundations of human cognitive biology [2,5,6]. Moral dilemmas arise when two or more of these principles come into conflict [1,7].

## Moral intuitions

Moral intuitions have been of scientific interest for decades. Several studies, including controlled and hypothetical scenarios, have been conducted to investigate this topic, wherein participants must make decisions based on their principles and ethical guidelines. The most common dilemma is the trolley problem, developed in the 1960s [3] And first proposed by Philippa Foot in 1967 as an example to discuss the applicability of the Double Effect Doctrine and consequentialism regarding when it is permissible to sacrifice a person [4].

Judith Thomson later took up this example of the trolley dilemma. Variations of the original problem were introduced, allowing discussion of the morally relevant differences in each case and why people preferred one option over another when the outcomes were the same across all variations. [8]. Although the variations of the trolley problem mentioned situations involving boys and girls, most research has focused on describing moral thinking in adult populations. There are currently very few studies focusing on the child population. These studies have addressed utilitarian preferences between children and adults [9,10], and the moral judgments on which boys and girls base their decisions on the cost-benefit of each case [11].

On the other hand, John Mikhail et al. have studied the application and consistency of the UMG and its three deontological principles. The UMG has been used to evaluate 200,000 participants across over 120 countries, with results demonstrating that people have innate, implicit knowledge of legal and moral principles that they automatically apply when making decisions in moral cases [1,12–14]. In addition to the three principles, several deontological rules are thought to integrate people's morality.

## Moral intuitions vs moral judgment

On the other hand, Piaget and Kohlberg's theory has demonstrated that children can understand concepts such as justice, fairness, and equality and make moral judgments. However, the argumentative justification for moral dilemma decisions differs from that found in studies using trolley scenarios to examine moral intuitions. According to Kohlberg, for example, human morality may have a considerable social influence [15–18]. At the same time, the trolley scenario studies seek moral intuitions supported by UMG, defined as "spontaneous ad hoc reactions about the moral

quality of a certain action or the omission of this action in a moral dilemma" [1]. However, they allow room for exploration of the influences in morality, although the elements needed for individual variations of moral decisions made by people are not defined [1,11,12].

Considering both moral intuitions and reflected moral judgment, Greene formulated "The Dual Process Model of moral judgment", where it is mentioned that there are two processes in how we make moral judgments. On the one hand, we have automatic emotional reactions (moral intuitions), and on the other, a slower cognitive process. However, other authors [3,9] have mentioned that the first concerns judgments based on rules (deontology), and the second, utilitarian judgment. In his theory, Greene notes that this is not necessarily the case and that both processes can be found in both deontology and consequentialism [19]. Ng et al., through three experiments with 917 participants in total, found that those who reflected on reasons for specific moral dilemmas showed a consistent increase in sensitivity to norms, but not in sensitivity to consequences, that is, when reflecting there is a preference for deontological principles, thus casting doubt on whether these principles are only carried out through moral intuitions [20].

### Research problem

In 2019, Niklas Dworazik et al. employed trolley scenarios to compare moral intuitions between children and their mothers, and to explore their deontological principles. They studied 56 dyads of children aged 3–6 years and their mothers. They found an overall agreement of 71.1% for the five dilemmas between children and their mothers, which was higher than the agreement by chance (57.5%) found by a permutation test (see statistical analysis) (z = 5.56, p < 0.001). Likewise, they found significant correlations (r = 0.336–0.404) when evaluating the dilemmas separately. The research concluded that children responded to moral dilemmas like their mothers, and that the family context might be a critical factor in explaining inter-individual variation in their responses to ethical dilemmas [3].

To understand how people shape their ethical judgments, it's essential to study the subject from the earliest stages of life, exploring the factors that contribute to ethical judgment development, such as family. It is unknown whether the level of agreement between children and their mothers varies significantly between populations with different sociodemographic characteristics. To our knowledge, it is also unknown whether the family situation resulting from the child's illness could influence the decisions made in ethical dilemmas and the concordance between mothers and children.

### Research aim

For this reason, it is essential to study this subject in a different population than in Niklas Dworazik´s studies, thereby determining the influence that primary caregivers can exert on children's judgments in other cultures. This study aims to evaluate the agreement between Mexican mothers and their preschool children when responding to moral dilemmas.

## Materials and methods

### Ethical considerations

Research and Ethics Committee of the Federico Gómez Children's Hospital of Mexico, with the approval of protocol HIM-SR-2021–003 dated July 21, 2021.Written informed consent was obtained for all child/mother dyads included in the study.

The period was from August 1, 2021, to March 30, 2022. The consent informed participants about the purpose of the research and the study's application procedure. Neither direct nor indirect financial remuneration was given to any participant. A copy of the signed consent was provided upon completion of the test.

In this case, preschool children are considered a vulnerable population. Therefore, informed consent was requested from their parents or guardians. To avoid the negative impact that presenting a moral dilemma could have on the child, these were specifically designed so that the outcome would not be death, but only a child injured by a ball.

## Study design

An analytical cross-sectional study was conducted, with a quantitative approach, based on the original research by Niklas Dworazik [3], who developed an instrument to evaluate responses to moral dilemmas in children and adults.

## Participants

Using convenience sampling [21], 85 patients and their mothers (dyads) attending the outpatient clinic of a third-level pediatric hospital were invited to participate with children aged 3–6 years who were accompanying their mothers. Three participants with a diagnosis of cognitive delay or mental disability were excluded, and the remaining 82 dyads (mother and child) answered the five dilemmas above, as well as the two control dilemmas, to ensure understanding and conscious responses. Seven tests that were not completed correctly were discarded. In six of these, the children responded, but the mothers were unable to answer because it was time for their medical appointments. They were not located again. In the seventh, the mother had a family emergency, so she had to leave the hospital without having finished the test. The responses of the 75 remaining dyads were analyzed.

## Sample size

Considering the study by Dworazik [3]. Sample size was calculated with a formula for correlation tests [22]. The highest correlation coefficient from previous research (0.36) was used, with a bilateral significance level of 0.05 and a statistical power of 80%, yielding a total of 62 subjects. However, it was possible to recruit a bigger sample.

## Instrument description

The instrument consists of five cases to be assessed and two control cases, ensuring that stories are correctly understood (see Table 1). It was designed to be administered using folders that contained pictures accompanied by brief sentences describing the situation, and the answers to each case were dichotomous (whether the participant would take action or not). To avoid bias due to the gender of the participant, the characters in the stories can be male or female depending on each child's gender, i.e., if the participant is a girl, the character will be a girl, and if the participant is a boy, the story character will be a boy. In the case of the mothers, the characters were female. The following are the fundamentals of each case presented to mothers and children:

- The original trolley idea, the principles of omission and intention, support the Bystander case. In this case, the character must divert a ball onto a secondary path to save five children, risking one person's life.

- The Footbridge case is based on a variation of the trolley problem. It also captures the principles of contact and intention as the character must push a child with a large backpack over a bridge to stop a ball and save the other five children, although the child with the backpack will be injured by the fall and the impact of the ball.

- The Expensive Equipment case is similar to the Bystander case; however, instead of choosing between hurting a person or a group, the decision entails choosing between injuring a person or a set of toys, and the principle of omission also supports it. The Implied Consent case presents a child being threatened by a ball rolling at high speed. The character can push the child to prevent the ball from hurting them, but the child would be slightly injured. As with previous cases, this case is based on the principle of intention.

- The Drop Man case is based on the Contact Principle because, unlike the Footbridge case, the character can prevent the ball from hurting the children's group by activating a lever that would let the child with the backpack fall, bearing the damage caused by the fall and the impact of the ball.

**Table 1. Moral dilemmas: case description.**

| Moral Dilemmas | |
| --- | --- |
| Bystander | A ball threatens a group of five children. The character can redirect the ball to an alternative path, thus saving the group from being hit, but hurting one child in the process. |
| Footbridge | Unlike the Bystander case, the only way to save the five children is by pushing a child with a big backpack over the bridge. Thus, the child hits and stops the ball, but also gets hurt in the process. |
| Drop Man | Unlike the Footbridge case, the only way to save the children is by remotely activating a hatch through which the child with the backpack falls, intersecting the ball's path. |
| Implied Consent | In this case, only one child is threatened by the ball. The character may push the child out of the ball's path, saving them from severe injury, though the child would still be slightly hurt. |
| Expensive Equipment | The only way to save the set is to redirect the ball to an alternative path that would not harm a child. |
| Control Dilemmas | |
| Costless Rescue | This control case is similar to the Bystander case. However, since no child is on the alternative path, redirecting the ball would cause no harm. |
| Disproportional Death | In this control case, the ball threatens only one child. Redirecting the ball to an alternative path would hurt the other five children. |

*Note.* Dilemmas adapted from: [3].

## Case-control scenarios

- The first control case, Disproportional Death, is based on the Bystander case and only reverses the position of the group and the person standing alone. In this case, redirecting the ball would now hurt the group of five children.

- The second control case, Costless Rescue, is based on the original trolley idea. Still, in the secondary path, no object or person will get hurt if the character chooses to divert the ball from the fence.

## Application procedure

Every mother-child pair in the waiting room of each outpatient specialty clinic was invited and encouraged to participate in the study. The test application took approximately 15 minutes to complete. The same dilemmas were applied simultaneously but individually to each mother and child by two trained applicators, who had folders with pictures of the dilemmas. Each applicant performed a test: one for the child, who read each dilemma as a story. The other applicator evaluated the mother using another test that included a description of the dilemma and images to reinforce the information. Mothers and children were in separate areas within the waiting room, so they could neither hear nor see each other. Because the children were young, before separating them from their mothers, the procedure was explained to them, and where their mothers would be waiting for them as they responded to the interviewers. They answered five moral dilemmas and two control cases. After each question, they were asked whether they should act on the dilemma (with options of action or omission).

The applicants were instructed and trained to conduct the applications systematically and consistently. They were only to read the established dialogues in a neutral style to avoid influencing responses to the dilemmas. The characters' names in the stories were changed to typical Mexican names to make them friendlier to the participants.

Two control questions ("Can the children see the ball roll down the hill?" and "Can the child see the ball roll down the hill?") were asked of the children and their mothers to ensure understanding of the dilemmas and that the answers were answered consciously.

Two answer sheets were used, one for the mother and another for the child to indicate their answers to the applicator, who recorded them. Likewise, both the child's characteristics (age, gender, and school attendance) and the mother's (age, place of residence, monthly income, employment status, educational background, and marital status) were recorded. Subsequently, all the data obtained were entered into an Excel database by one of the researchers to clean the data and ensure accuracy and precision. Finally, the data were exported to SPSS v.24 for statistical analysis.

## Statistical analysis

Descriptive statistics were used to describe sociodemographic variables and the results from the dilemma tests. Qualitative variables were reported as counts and percentages, and quantitative variables were reported as means and standard deviations. A permutation test assessed the overall agreement between mothers' and children's responses across the five dilemmas. The agreement distribution obtained in the actual sample was compared with that from the permutation test (a random distribution with 1,000 samples) using a z-test. Thus, the sample distribution was compared to a random distribution to dismiss the random effect. Kappa Statistic and Spearman's correlation analyses were conducted to assess agreement between mothers and children for each dilemma [23]. Binomial tests were performed at the 50% chance level for each dilemma, with children and mothers analyzed separately to control for the random effect. To determine the Double Effect Doctrine, McNemar's test was used to compare the Bystander vs. Drop Man dilemmas and the Footbridge vs. Drop Man dilemmas to assess the presence of the Contact Principle [3]. The statistical significance level was set to <0.05.

The permutation test is a non-parametric statistical technique that allows us to make inferences about our data without assuming a specific distribution. It is handy in cases of non-normal data or small samples, where traditional methods may not be adequate. Through resampling, we can construct an empirical distribution of the statistics of interest and evaluate the variability of our estimates without assuming a specific distribution [24].

This evaluates whether the statistics from the original sample differ significantly from those of the empirical distribution. As such, the p-value represents the proportion of times that the value of the calculated statistic is equal to or greater than the observed statistic. Therefore, a p-value less than 0.05 indicates a statistically significant difference between groups. In our case, a permutation test with 1,000 resampling was used [25].

## Results

The sociodemographic features of the 75 dyads are described in Table 2. Notably, 60% of children were boys, with an average age of 4.4 years, and 60% were enrolled in school. The average age of mothers was 33 years; 32% had a job, 76% had a high school or technical school degree as their highest level of education, and 73% lived with a partner.

### Answers to dilemmas

Answers to each dilemma can be found in Fig 1. Binomial tests, conducted at a 50% chance level for each dilemma, showed that both mothers and children tended to intervene in the Bystander dilemma ($p < 0.01$). Mothers preferred not to take action in the Footbridge and Expensive Equipment dilemmas ($p < 0.001$) as well as in the Implied Consent case ($p = 0.037$). A clear answer to the Drop Man case was not found ($p = 0.1$). Except for the Bystander dilemma, children's responses did not show a clear tendency in all other dilemmas ($p > 0.06$). Fig 1B shows the differences between mothers and children when solving each dilemma. There were no significant differences, as determined by the $X^2$ test.

The Double Effect Doctrine was evaluated by comparing the Drop Man and the Bystander dilemmas. Mother's answers revealed that they followed the doctrine (McNemar = 13.04, $p < 0.001$). When comparing the Drop Man and the Footbridge dilemmas, the mothers' answers indicated that they also followed the Contact Principle (McNemar = 1.02, $p < 0.001$).

**Table 2.  Mother–Child sociodemographic features.**

| Family Features | n | % |
|---|---|---|
| Residence | | |
| Mexico City | 37 | 49.3 |
| Other Mexican cities | 38 | 50.7 |
| Income | | |
| <10,000 | 68 | 90.7 |
| 10,000–20,000 | 7 | 9.3 |
| Patients' Features | | |
| Age | | |
| 3 years old | 19 | 25.3 |
| 4 years old | 22 | 29.3 |
| 5 years old | 17 | 22.7 |
| 6 years old | 17 | 22.7 |
| Gender | | |
| Female | 30 | 40 |
| Educational attainment | | |
| Not attending school | 30 | 40 |
| Pre-school | 39 | 52 |
| Primary school | 6 | 8 |
| Mothers' features | | |
| Employment Situation | | |
| Employed | 24 | 32 |
| Unemployed | 51 | 68 |
| Educational Level | | |
| Elementary school | 7 | 9.3 |
| Middle school | 28 | 37.3 |
| High school/Technical school | 29 | 38.7 |
| Bachelor's degree | 8 | 10.7 |
| Postgraduate | 3 | 4 |
| Marital Status | | |
| Single | 16 | 21.3 |
| Consensual union | 26 | 34.7 |
| Married | 29 | 38.7 |
| Divorced | 4 | 5.3 |

*Note. N*=75. Patients were, on average, 4.4 years old (*SD* = 1.1). Mothers were, on average, 33.4 years old (*SD*=8.5).

Regarding children, the comparison revealed that they neither follow the Contact Principle (McNemar = 11.9, p = 0.21) nor the Double Effect Doctrine (McNemar = 0.35, p = 0.23).

Regarding the correlation between children's answers and their age, older children showed a greater tendency not to intervene in the Expensive Equipment case (Spearman's rho = 0.28, p = 0.014). No significant correlations were observed in any of the other dilemmas.

## Differences between boys and girls

The mothers, boys, and girls were divided into groups to evaluate if there were differences between the groups (see Table 3) in the frequency with which each group answered that they would take action in the three central dilemmas comprising UMG

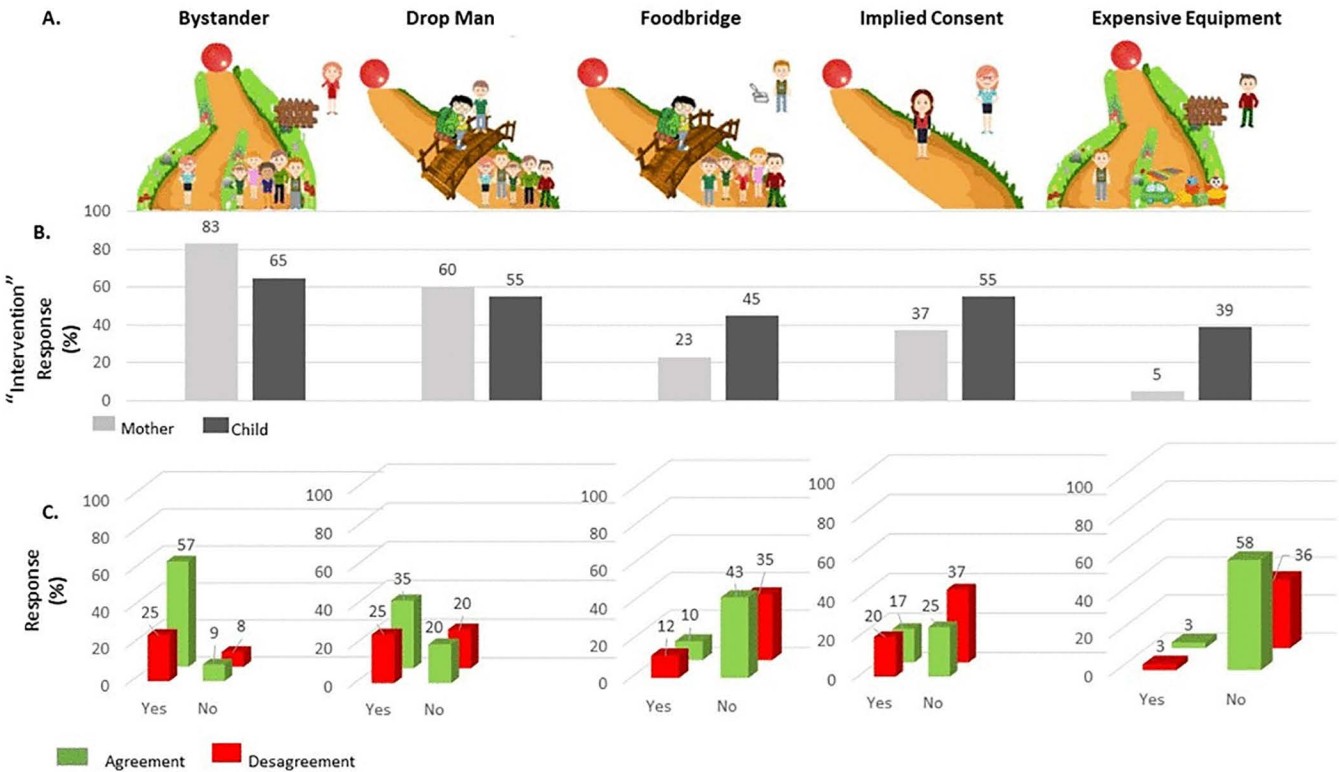

A.- The five scenarios
B.- Responses of the five scenarios. The bars represent the percentage of YES responses to the intervention
C.- Correspondence between maternal and child response by intervention (yes) and by omission (no). Agrrements (green) and disagreementes (red) of the mother-child dyads

**Fig 1. Answers to dilemmas.**

(Bystander, Footbridge, and Drop Man). Results showed that the boys acted similarly to the mothers, choosing to act in the Bystander dilemma. The girls, on the other hand, acted similarly to the mothers in not taking action in the Footbridge dilemma. Similarly, unlike the girls, a slight trend was found in the boys for acting in the three dilemmas.

## Mother–child agreements

Fig 1C shows the mother-child agreement pattern of each dilemma. A Global Agreement (GA) ≥55% was observed in three out of five dilemmas, although a statistical significance with Kappa and Spearman's correlation analyses could

**Table 3. Differences between boys and girls in the three central UMG dilemmas.**

| Dilemma | Mothers | | Girls and Boys | | Girls | | Boys | |
|---|---|---|---|---|---|---|---|---|
| | *n* | % | *n* | % | *n* | % | *n* | % |
| Bystander[a] | 62 | 83 | 49 | 65 | 16 | 53 | 33 | 73 |
| Drop Man[a] | 45 | 60 | 41 | 55 | 14 | 47 | 27 | 60 |
| Footbridge[a] | 17 | 23 | 34 | 45 | 11 | 37 | 23 | 51 |

*Note.* For mothers *N* = 75, for girls and boys *N* = 75, for girls *N* = 30, for boys *N* = 45.

[a] Reflects the number and percentage of participants who choose to act in each dilemma.

not be found: Bystander (GA=67%; κ=0.16, p=0.11; Spearman=0.18, p=0.11); Drop Man (GA=55%; κ=0.07, p=0.5; Spearman=0.07, p=0.5); and Expensive Equipment (GA=61%; κ=0.03, p=0.6; Spearman=0.05, p=0.6). However, when considering the overall answers to the five dilemmas, the GA was 56%, which was higher than the GA observed in the random answers pattern (GA: 48%; κ=2.19, p=0.03) (corresponding to 2.19 standard deviations from the swapped distribution average). Please refer to the Statistical Analysis section for more details.

## Control cases

As for the control cases, children and mothers are to take action in the Costless Rescue case (see Fig 2B). While children showed no clear trend in whether to take action in the Disproportional Death case, mothers did prefer to intervene (see Fig 2B). No significant agreement was found between mothers and children in these control cases (see Fig 2C).

As a secondary objective, we compared our results with the results of the study by Dworazik et al (Table 4).

## Discussion

This study revealed a significant global agreement between mothers and children regarding the five dilemmas. This was observed during the permutation analysis, where a higher global agreement degree (56%) was observed than for randomly assigned dyads (48%). However, unlike what Dworazik et al. [3] found, the agreements between mother and child were lost when analyzing dilemma by dilemma. One explanation is that our results may have been influenced by the fact that our children were patients at a tertiary-level hospital and most had been diagnosed with a chronic disease, unlike the German study, which included children from nurseries. It has been described that chronic diseases in children are associated with mental health problems in adult life, such as depression, anxiety, and aggression, and have been associated with academic and social issues during childhood and adolescence [26,27]. This does not necessarily mean that the moral development of children is affected by having a chronic disease; however, the findings of this study raise the question for future research of whether the presence of a chronic disease is associated with how children respond to moral dilemmas. Another explanation may relate to socioeconomic status. Though we did not specifically measure this variable in our population, most outpatients at this hospital have low socioeconomic status, unlike Dworazik's research, which studied dyads in an urban middle-class context [3]. A 2022 study by Peretz-Lang et al. found that socioeconomic status influences preschoolers' moral judgments regarding the distribution of resources [28]. The finding that the Contact Principle was absent in both German and Mexican participants may be explained by the fact that cognition is still developing in preschoolers. Evidence shows that utilitarian moral judgments are cognitively more demanding and thus develop later in life [29].

Compared with the German study, we found a trend toward action across the four groups. This trend is more pronounced in the Bystander dilemma, slightly decreases in the Drop Man dilemma, and dramatically decreases in the Footbridge dilemma, indicating a trend to consider the ethical principles established in the Double Effect Doctrine and the Contact Principle. This is clearer in German and Mexican mothers, where both principles are significant. We did not find differences in our children population when comparing the Drop Man and Bystander dilemmas regarding the Double Effect Doctrine and the Drop Man and Footbridge dilemmas regarding the Contact Principle. Hence, none of the principles were considerably present in our children, unlike the German study, where children were found to apply the Double Effect Doctrine but did not make moral judgments according to the Contact Principle [3].

As we know from previous studies [1,3,13], the Double Effect Doctrine and the Contact Principle are present in most adults and, without a doubt, the findings found in this study in Mexican preschoolers, in whom the Doctrine of the Double Effect is not yet present, unlike the German preschools where it is present, will give rise to future studies that evaluate which are the factors that influence these differences found when using these principles, being potential factors to study the culture, socioeconomic and health status of the participants.

This study showed that mothers and children prefer to take action in the Bystander dilemma. Conversely, both preferred not to act in the Expensive Equipment dilemma, indicating that most children and mothers prioritize saving a human

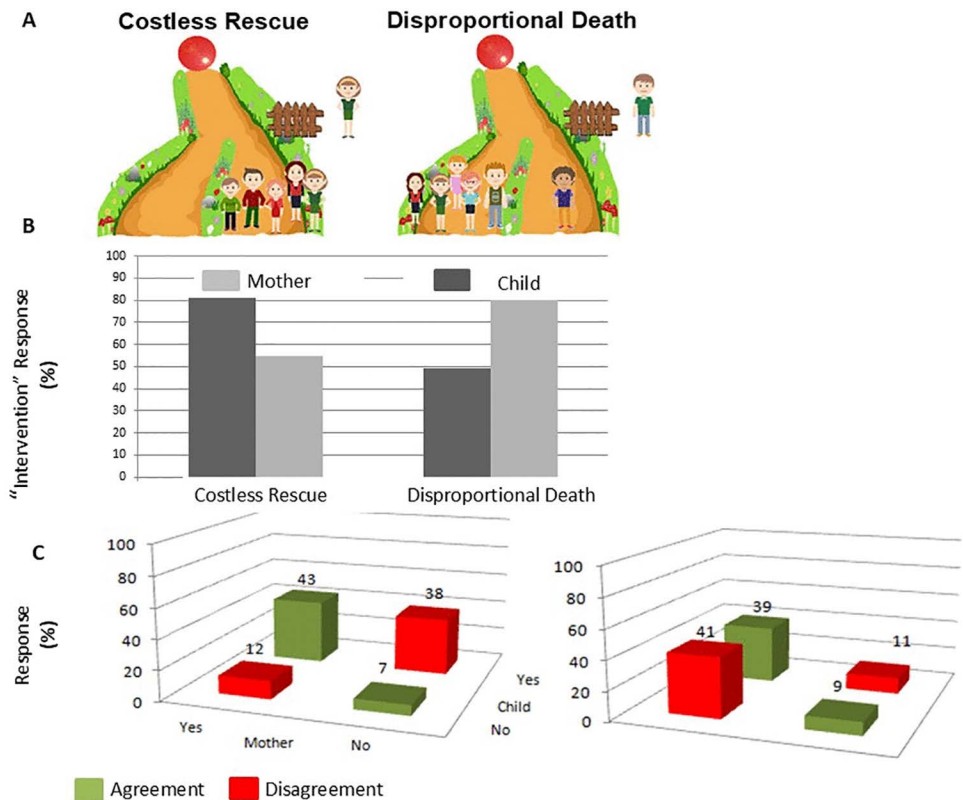

A.- The two control scenarios
B.- Responses of the two control scenarios. The bars represent the percentage of YES responses to the intervention
C.-Correspondence between maternal and child response by intervention (yes) and by omission (no). Agreements (green) and disagreements (red) of the mother-son dyads

**Fig 2. Control cases.**

**Table 4. Differences in answers to dilemmas in the German and Mexican studies.**

| Dilemmas | Mothers | | | | Children | | | |
|---|---|---|---|---|---|---|---|---|
| | Germany | | Mexico | | Germany | | Mexico | |
| | *n* | % | *n* | % | *n* | % | *n* | % |
| Bystander[a] | 44 | 79 | 62 | 83 | 43 | 77 | 49 | 65 |
| Drop Man[a] | 20 | 36 | 45 | 60 | 32 | 57 | 41 | 55 |
| Footbridge[a] | 10 | 18 | 17 | 23 | 31 | 55 | 34 | 45 |
| Implied Consent | 46 | 82 | 28 | 37 | 42 | 75 | 41 | 55 |
| Expensive Equipment | 5 | 8.9 | 4 | 5 | 21 | 37 | 29 | 39 |

*Note.* For Germany *N*=56, for Mexico *N*=75.

[a] Reflects the number and percentage of participants who choose to take action in each dilemma.

life over an object, regardless of its high value. However, like in the German study, just over one-third of the children decided to save the toys, suggesting that preschoolers may still be developing their understanding of the moral value of a

person. For future research, it would be interesting to investigate the age at which individuals begin to practice deontological principles and understand moral values, as well as the factors involved in moral development.

We consider that the combination of innate and environmental factors influences the development of children's morality. Some studies have linked genetic variations to different types of moral judgments [30] and to antisocial behavior, which are considered moral deviations [31,32]. Conversely, other authors emphasize environmental factors; for example, in their 2021 systematic review, Costa Martins et al. found a relationship between secure attachment and improved moral development in children, particularly in moral emotions such as empathy [31], a factor known to be essential for the development of moral reasoning. Although theoreticians of moral reasoning development and some socio-cognitive research highlight that moral judgments are universal, the influence of socio-cultural context on people's moral development cannot be denied [32]. Therefore, differences between the German and Mexican cultures may have influenced the study results. A potential avenue for future research would be to evaluate all the factors involved in children´s moral development.

## Theoretical implications

The results of this study provide relevant evidence for the debate on Universal Moral Grammar (UMG) and the development of deontological principles in childhood. The absence of the Doctrine of Double Effect and the Principle of Contact in Mexican preschool children, even when their mothers do apply them, suggests that these principles are not internalized in the early stages of moral development but require a longer process of cognitive and emotional maturation. This not only reinforces positions such as Greene's double-process model [19], which states that moral judgments are shaped by the interaction between rapid emotional intuitions and slower reflective processes that develop differently across age, but also aligns with Jonathan Haidt's theory of Moral Foundations [33,34]. According to this theory, there are innate moral foundations such as "care/harm" and "justice/deception" that emerge early (related to empathy, care, justice) while others, such as authority, loyalty, or purity, develop later, influenced by social, cultural, and family norms. Recent studies tracking the use of moral language in childhood find that terms associated with care and justice appear in the early years. In contrast, those associated with authority and purity emerge more strongly later as children gain greater social and cognitive exposure [35]. All of this suggests that the most abstract deontological principles are likely to require not only advanced cognitive abilities (theory of mind, inhibition, moral reasoning) but also a certain degree of structured moral socialization, which aligns with this study's findings.

Likewise, the comparison with the German study [3] shows that cultural, health, and socioeconomic factors can modulate the early expression of moral principles, which partly questions the strictly universal character of the UMG [7,31]. In this way, the theory must integrate not only innate biological bases but also the influence of family and cultural context on the acquisition and manifestation of moral principles.

Finally, the findings that both children and mothers prioritize saving human lives over material goods confirm the existence of a shared core of morality from an early age, suggesting that universal predispositions may shape certain aspects of morality. In contrast, others, more complex, depend on socialization and cultural learning processes.

## Practical implications

In practical terms, these results have several applications:

Moral education in childhood: International evidence suggests that structured school programs can strengthen key components of early moral judgment (empathy, self-regulation, deliberation, and sensitivity to norms). For example, the PATHS (Promoting Alternative Thinking Strategies) program showed positive effects on children's socio-emotional competence in cluster-randomized trials and implementation studies in European and Anglo-Saxon contexts. However, the effects vary depending on fidelity and implementation context. [36,37]. Randomized clinical trials evaluating programs explicitly focused on empathy, such as Roots of Empathy, have shown reductions in aggression and bullying

and increased prosocial behaviors in children. [38,39] The Second Step (Social-Emotional Learning) program has also demonstrated improvements in socio-emotional skills and reductions in disruptive behaviors in controlled studies; however, results are heterogeneous across contexts and depend on the quality of implementation, socioeconomic level, and schooling. [40,41].

Taking into account the available evidence, interventions aimed at enhancing moral reflection in preschool and early grades could be developed and evaluated, including: a) age-appropriate content that works on empathy and perspective (via stories, games, and experiential activities); b) training elements for teachers and families that guarantee fidelity and continuity; and c) systematic evaluation of processes (not only results) to identify which components of the program facilitate the development of more complex deontological reasoning.

Support for families with children with chronic diseases: Given that the sample included children in follow-up in a tertiary hospital, where most of them have a chronic disease, the possibility opens that the experience of the disease influences the way in which children process moral dilemmas. This highlights the importance of psychological and educational interventions that take into account the ethical and emotional dimensions of this vulnerable group.

Public policies and parental training: The results show that mothers already incorporate more elaborate moral principles than their children do. This implies that parental support strategies could be aimed at strengthening parenting practices that foster empathy, moral reflection, and exposure to ethical decision-making contexts at home.

Intercultural implications: The differences observed between Mexican and German children suggest that ethics training programs should not assume cultural homogeneity but rather recognize the influence of socioeconomic and cultural contexts on moral development, thereby promoting educational policies sensitive to diversity.

## Limitations of the study and future research

This study has several limitations that should be considered when interpreting the results. First, the sample size was relatively small (n = 75), which may limit the generalizability of the findings to other child and maternal populations, especially in different socioeconomic and cultural contexts. Although a robust statistical analysis (a permutation test) was used, studies with larger, more representative samples would further confirm the results' robustness. On the other hand, since the presence of the interviewer may influence the participants' responses by wanting to give socially accepted answers or being ashamed [42,43]. We attempted to mitigate this bias by using a standardized test and by employing qualified, trained applicators.

Third, the sample was composed of children treated in a tertiary hospital, many of whom had a history of chronic diseases. This may have influenced their responses, as the literature has documented that childhood health conditions can affect emotional and social development [26,27]. Therefore, future studies should include clinical and non-clinical populations to differentiate the effect of the disease on moral reasoning.

Fourthly, the cross-sectional design used does not allow for the establishment of development trajectories. Child morality is a dynamic process that changes with age, so it would be advisable to conduct longitudinal studies to observe the evolution in the application of deontological principles and the influence of family, educational, and cultural factors at different developmental stages.

Another significant limitation is the absence of qualitative measures that examine the justifications and reasoning behind children's and mothers' responses. The quantitative approach used allowed detection of patterns of response, but not the reasons underlying these decisions. Future research could benefit from mixed methods approaches that combine structured dilemmas with interviews or focus groups to explore in greater depth the cognitive and emotional processes involved.

Finally, the study did not directly assess key contextual variables such as socioeconomic status, religiosity, parenting style, or exposure to specific social norms, which could mediate the relationship between mothers' and children's responses. Including these factors in future research would allow us to build a more comprehensive model of moral development in childhood.

## Conclusions

To conclude, although preschoolers and their mothers share certain similarities in responding to moral dilemmas, the children in this study did not integrate deontological principles into their responses, which makes their responses different from those of their mothers. On the other hand, mothers use these principles, corroborating the previous evidence. For future research, it would be interesting to evaluate older children to determine at what age they begin to apply these principles and to assess the innate and environmental factors that may influence their moral development.

## Supporting information

**S1 File. "DATABASE Moral Dilemmas": This database contains confidential data that allows the replication of the results of this research.**
(XLSX)

## Author contributions

**Conceptualization:** María G. Jean-Tron, Juan Garduño-Espinosa, Onofre Muñoz-Hernández.

**Formal analysis:** Guillermo Salinas-Escudero.

**Investigation:** Diana Ávila-Montiel.

**Methodology:** María G. Jean-Tron, Guillermo Salinas-Escudero.

**Project administration:** Gina C. Chapa-Koloffon, Oscar A. Resendez-Berber, Elizabeth Cruz Cruz.

**Supervision:** Gina C. Chapa-Koloffon, Oscar A. Resendez-Berber, Elizabeth Cruz Cruz.

**Validation:** Gina C. Chapa-Koloffon, Oscar A. Resendez-Berber, Elizabeth Cruz Cruz.

**Writing – original draft:** María G. Jean-Tron, Juan Garduño-Espinosa, Guillermo Salinas-Escudero.

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
