## [Decision Letter · Decision Letter 0]

12 Jun 2025

Dear Dr. Garduño-Espinosa,

Thank you for submitting your manuscript to PLOS ONE. After careful consideration, we feel that it has merit but does not fully meet PLOS ONE’s publication criteria as it currently stands. Therefore, we invite you to submit a revised version of the manuscript that addresses the points raised during the review process.

The manuscript has been evaluated by two reviewers, and their comments are available below. The reviewers have raised a number of concerns that require attention. Specifically, they noted issues related to claims not fully supported by the data, inadequate methodological descriptions, and insufficient alignment between the theoretical framework and the analysis. In addition, the reviewers requested more details to clarify and justify the methodological approach and to ensure that interpretations of the findings remain closely tied to the data presented.

Could you please carefully revise the manuscript to address all comments raised?

We look forward to receiving your revised manuscript.

Kind regards,

Zahra Al-Khateeb, Ph.D

Staff Editor

PLOS ONE

Journal Requirements:

Reviewers' comments:

Reviewer's Responses to Questions

**Comments to the Author**

1. Is the manuscript technically sound, and do the data support the conclusions?

Reviewer #1: Partly

Reviewer #2: Partly

2. Has the statistical analysis been performed appropriately and rigorously?

Reviewer #1: No

Reviewer #2: I Don't Know

3. Have the authors made all data underlying the findings in their manuscript fully available?

Reviewer #1: No

Reviewer #2: Yes

4. Is the manuscript presented in an intelligible fashion and written in standard English?

Reviewer #1: No

Reviewer #2: Yes

Reviewer #1: In general, the topic itself suggests a qualitative methodological approach. But if it is still going to be based on statistics (which is very doubtful in this case), then the whole methodology part needs more substantiation and proof from the previous studies and sources, as well as sample expansion. Major revisions are vital.

Reviewer #2: This is on the whole a well written piece describing a small scale experiment that builds on prior experiments. The only part I find objectionable is the claim on page 14 that "This study showed that mothers and children prefer to take action in the Bystander dilemma, displaying a preference for utilitarianism." I think the second part of this claim regarding the 'preference for utilitarianism' is neither supported by the data or needed for the main claims of the paper and should be removed.

There are some other more minor issues and points to be addressed. Please find some further comments by page number below to assist you in your revisions.

2: “feasible” Should this be ‘permissible’?

3: Describe the permutation test.

“The research concluded that children responded to moral dilemmas similar to their mothers” Was it the dilemmas that were similar or their response that was similar? Perhaps ‘similarly’ or ‘in a similar way to’.

4: “To avoid bias due to the sex of the participant, the characters in the stories can be male or female depending on each child’s gender, i.e., if the participant is a girl, the character will be a girl, and if the participant is a boy, the story character will be a boy.” How was this kind of bias controlled for in the case of the mothers. Address this point somehow.

6: “Seven tests that were not completed correctly were discarded”. Explain further. What were the errors?

7: “Neither financial remuneration nor any other kind of retribution”. Is “retribution” what was intended here?

“Mothers and children were in separate areas within the waiting room, so they could not hear each other's answers.” Could they see each other? Could this have an effect?

8:

“Two control questions were asked to ensure understanding of the dilemmas and that the answers were answered consciously: "Can the children see the ball roll down the hill?" and "Can the child see the ball roll down the hill?"” Asking the question of negative examples, as is common in ‘theory of mind’ tests, would probably still yield some affirmative answers. Address this point.

13: “how children respond to moral dilemmas Another” Missing full stop.

14: “This study showed that mothers and children prefer to take action in the Bystander dilemma, displaying a preference for utilitarianism.” Why does this show this preference? I am not sure that it does. I think the claim goes beyond the findings of the experiment. There are surely other ethical theories that licence the same binary action, or ways to argue that they would. How have they been excluded? What literature is being followed here? Who would disagree? Further, if the test had been conducted in 500 BCE (or at least a time before the theory of ‘utilitarianism’ was made explicit) what would the results have shown in that case? Surely not a preference for utilitarianism.

15: “However, like in the German study, just over one-third of the children decided to save the toys, so preschoolers may still be developing their understanding of how important people’s moral values are.” ‘how important the moral value of a person is’ It is not their ‘values’ that are at stake.

“Future research will be interesting if all the factors involved in children’s moral development are evaluated.” Rephrase, for example: ‘A potential avenue for future research would be to evaluate all of the factors involved in children’s moral development.’

“be fully generalizable on the other hand” Run-on sentence.

“which makes them different from their mothers' responses.” ‘which makes their responses different’

**Do you want your identity to be public for this peer review?** For information about this choice, including consent withdrawal, please see our Privacy Policy

Reviewer #1: **Yes:** Dr. Inga Iždonaitė-Medžiūnienė

Reviewer #2: No

---

## [Author Response · Author response to Decision Letter 1]

21 Jul 2025

We believe that the manuscript has been improved, and the final result is more precise and easier to read. We are grateful to the reviewers for their detailed and helpful comments.

Reviewer 1

The topic is very significant and up to date in the world that was so much affected by pandemics. However, there is a lack of need and significance substantiation of this research (in this paper).

The title of the article does not seem to clearly reflect the whole idea, and it is more of a sentence rather than a title. The description of deontological principles should also have a place in the data analysis section, showing the principles’ connection with the emerged dilemmas.

Answer: Following your suggestion, we revised the title to "Early Ethics: Exploring Moral Intuition and Maternal Influence in Preschool Children," which we consider substantially more compelling than the previous version.

The Introduction describes the three main deontological principles of the Universal Moral Grammar (UMG). In the description of the instrument, the scenarios for each dilemma and their relationship with deontological principles are outlined. For example, in the case of the Bystander scenario, the principle of action is evident, and in the case of the Footbridge scenario, the principle of contact is clear. In the statistical analysis section, it is now noted which scenarios were used to evaluate the Doctrine of Double Effect and the Contact Principle.

The small theoretical part seems very undeveloped and lacks more substantiation of moral dilemmas and their coherence with the deontological principles in data analysis. The analysis of the origin and reasons for the dilemmas that emerge would add value to the literature review. The problems parents experience at work should have an impact on moral dilemmas between mothers and children, as all experiences in certain environments may have various effects, especially in the case of destructive experiences. This topic is researched by the authors, which should be considered in the literature review part:

Answer: Thank you for the suggestions. A paragraph was added to the Introduction regarding the origin and reasons for moral dilemmas, as outlined in UMG, as well as Ng's findings and proposal, which contribute to the theoretical part of our study.

While the theoretical dilemma definitions were offered, there is a lack of clarity on the applicability of the results of this study. There is a lack of evidence on how the moral dilemma response similarity between mother and children’s dyads can be implicated in chronic disorders. Psychological disorders are not caused by one factor. It is a strong encouragement to the authors to make this interlink clearer by going back to the review of literature.

Also, consider the following questions for the review:

- How can moral dilemmas predict mental health challenges?

- What is the relationship between moral dilemma, congruence between mothers and children, AND mental health challenges?

Answer: While the objective of the study was not to investigate the influence of chronic diseases on responses to moral dilemmas, based on the found differences concerning the original research, we propose the hypothesis that by having a family member (in this case, the child) with a chronic disease, the family dynamics might change, and this could influence decision-making in moral dilemmas. In the discussion, we only provided an example of the effect that has been observed on mental health due to chronic diseases. In this case, there is evidence in the literature; however, there is no evidence of the effect on response to moral dilemmas. We believe that it could be a topic for future research.

First, there is a lack of a methodological approach. There is an ambiguity in the case research description: there is a need for clarification on research data collection and analysis methods in accordance with the selected methodological approach. Particularly, authors do not describe the data collection instruments, whether it is a survey or other, what sources these instruments were based on, etc. Later, the authors present tests, which lack a description of who created them and what the literature basis was for that creation.

Answer: This is a quantitative study, as clarified in the Methods section. By your kind suggestion, we included information about the authors and a description of the instrument used, which consists of five stories presented with drawings on sheets inside a folder. Each of the stories represents a moral dilemma that allows us to evaluate the concordance between mothers and children, as well as the presence of deontological principles.

The authors present a statistical analysis with a sample of 56, which is very small, and it is doubtful whether the results can be considered statistically significant. The other issue is that the authors did not substantiate the sample as reliable – literature sources should be used to prove that.

Answer: The sample size of our study was 75 dyads. Having based our study on the work by Niklas, which yielded significant results with a sample of 56 dyads, we consider that the 75 dyads reported would be sufficient. The reference is included in the manuscript.

Since a sensitive topic is being analysed, it is important to describe all necessary ethical issues for vulnerable groups and topics so that research ethics are not violated, especially as children (3-6 years) were separately researched. It should definitely be more elaborate. Children hearing about a trolley problem at that age are exposed to violence. If children were exposed to various topics such as the trolley problem, they would be exposed to latent violence by train. How was this addressed? Were there any safeguarding measures for children participating?

Answer: The scenarios are variations on classic dilemmas; they were designed precisely so that there would not be such explicit exposure to violence for young children. In none of the dilemmas are there any deaths, and the outcome is replaced by a child who is injured or in a similar situation. The dilemmas are described in the description section of the instrument. This is also clarified in the section on ethical considerations.

Test procedures were completed during the data collection. The need for more detailed clarification about tests is vital:

• Did the authors use two tests, or was it the same for mothers and children (even if children had stories read by trained persons)?

Answer: That's right, one test was used for the children, and a different test was used for the mothers; this is now clearly stated in the manuscript.

• Still, it is not clear how the data was collected – whether the participants had to write answers or tell them? Also, there is no relation to statistical analysis at this point. Were the participants asked some sort of questions? If so, what were they? This is before control questions.

Answer: The "Application procedure" section describes what the participants were asked and how their answers were obtained. The participants were asked if the protagonist of the story should act or not; the answers were recorded by the evaluator on an answer sheet that also contained data on the mother and child.

• Two control questions were asked, but the authors did not indicate who they asked to answer them – mothers or children.

Answer: It is now clarified in the manuscript that the control dilemmas were applied to both mothers and children.

The authors present the permutation test, ż test, the Kappa Statistic, Spearman’s correlation, and McNemar’s test in this part, but it is important to explain why they are used and give proof from other sources. The result part needs to be more detailed in explanations.

Answer: Information about the permutation test is added within the statistical analysis to justify its use and to increase its understanding. It is also clear what the McNemar test was used for.

Tables and figures are not usually used in the discussion part.

Answer: The table with the comparison of the two studies was placed in the results section

In the limitations section, the authors say that interviews were carried out. So, this is a huge misunderstanding of methodological approaches – interviews are qualitative methods and are not used in statistical analyses. Fifty-six samples are too small for statistics (even though it is convenience sampling); therefore, it is strongly recommended to think of a different and very clear methodological approach. In this part, it seems that trained applicators for both mothers and children are used, while in the methods part, it is different.

Answer: Yes, it lends itself to misunderstandings. We modified that part of the limitations since the evaluators did not conduct interviews but were trained to perform a standardized test. With this, we avoid the bias that could exist if we influence the participants' answers. Because the only data obtained through this instrument is whether the participant decides to act or not (a dichotomous response), we cannot conduct an analysis using a qualitative approach; for this, we would have to use interviews that allow us to delve deeper and collect other types of information. As the primary objective was to evaluate the agreement between the mother-child responses, we opted for a quantitative approach, which would also enable us to compare it with the German study.

The limitations part should be expanded, and further research issues should be discussed for practical and theoretical implications.

Answer: We expanded the limitations section. Throughout the discussion, future topics to be investigated are proposed, for example, whether the presence of a chronic disease, socioeconomic level, or culture can influence the responses to moral dilemmas; at what ages can the deontological principles described by the GMU be observed. Future research with a qualitative approach is needed to know the reasons or motives that children consider when solving moral dilemmas.

In general, the topic itself suggests a qualitative methodological approach. But if it is still going to be based on statistics (which is very doubtful in this case), then the whole methodology part needs more substantiation and proof from the previous studies and sources, as well as sample expansion. Major revisions are vital.

Answer: Thank you for the comment. We understand that a qualitative approach would allow us to address the topic in greater depth, which we will undoubtedly do in future research. However, we decided to adopt a quantitative approach to facilitate comparison with Niklas' previous work, using a methodology similar to theirs and a sample size comparable to theirs, albeit larger in our case. The reference to this study is included in the Methods section as the basis for the study design and methodology to be followed.

Likewise, information about the permutation test is included within the statistical analysis to justify its use and enhance its understanding.

Also, a more technical comment. The article needs further refining on the use of the English language. Please consider using language editing services to refine the manuscript. I.e., we cannot use shortened words “didn’t” for “did not” in academic writing. We typically use the full phrases “did not”, “cannot”, and so on.

Answer: Thank you; the manuscript has been revised and corrected.

Reviewer 2

This is, on the whole, a well-written piece describing a small-scale experiment that builds on prior experiments. The only part I find objectionable is the claim on page 14 that "This study showed that mothers and children prefer to take action in the Bystander dilemma, displaying a preference for utilitarianism." I think the second part of this claim regarding the 'preference for utilitarianism' is neither supported by the data nor needed for the main claims of the paper, and should be removed.

Answer: Upon reviewing the document, we consider your observation to be appropriate. To claim a preference for utilitarianism, we would need to know the arguments and justifications behind the answers to the dilemmas. Therefore, this statement was removed.

There are some other minor issues and points to be addressed. Please find some further comments by page number below to assist you in your revisions.

2: “feasible” Should this be ‘permissible’?

Answer: Thank you, this was corrected in the manuscript.

3: Describe the permutation test.

Answer: This is described in the statistical analysis section.

“The research concluded that children responded to moral dilemmas similar to their mothers.” Was it the dilemmas that were similar or their response that was similar? Perhaps ‘similarly’ or ‘in a similar way to’.

Answer: Thank you, the change has been made to the manuscript

4: “To avoid bias due to the sex of the participant, the characters in the stories can be male or female depending on each child’s gender, i.e., if the participant is a girl, the character will be a girl, and if the participant is a boy, the story character will be a boy.” How was this kind of bias controlled for in the case of the mothers? Address this point somehow.

Answer: Both in Niklas's original instrument and in ours, the stories used for mothers had female characters. It is corrected in the manuscript.

6: “Seven tests that were not completed correctly were discarded”. Explain further. What were the errors?

Answer: The tests were not completed mainly because they no longer had the time to do it, either because they had a medical appointment or because they had to leave the hospital relatively urgently. This information has been added to the participants section.

7: “Neither financial remuneration nor any other kind of retribution”. Is “retribution” what was intended here?

Answer: What we intended to say was that they were not given any economic incentive in exchange for their participation. The wording has been changed to refer to direct or indirect financial remuneration in the text.

“Mothers and children were in separate areas within the waiting room, so they could not hear each other's answers.” Could they see each other? Could this have an effect?

Answer: We tried to get the child to focus on the interviewers so that he wouldn't hear or see his mother. We consider that the answers could be altered if the child saw his mother.

13: “How children respond to moral dilemmas.” Another missing full stop.

Answer: Thank you, this has been corrected in the manuscript.

14: “This study showed that mothers and children prefer to take action in the Bystander dilemma, displaying a preference for utilitarianism.” Why does this show this preference? I am not sure that it does. I think the claim goes beyond the findings of the experiment. There are surely other ethical theories that license the same binary action, or ways to argue that they would. How have they been excluded? What literature is being followed here? Who would disagree? Further, if the test had been conducted in 500 BCE (or at least a time before the theory of ‘utilitarianism’ was made explicit), what would the results have shown in that case? Surely not a preference for utilitarianism.

Answer: Reviewing what is written in the document, we consider that your observation is appropriate, since to say that they have a preference for utilitarianism, we would need to know the arguments and justifications of the answers to the dilemmas, which is something that was not addressed in this research, so this statement was removed.

15: “However, like in the German study, just over one-third of the children decided to save the toys, so preschoolers may still be developing their understanding of how important people’s moral values are.” ‘How important the moral value of a person is.’ It is not their ‘values’ that are at stake.

Answer: Thank you, it was a drafting error, it was corrected in the manuscript

“Future research will be interesting if all the factors involved in children’s moral development are evaluated.” Rephrase, for example: A potential avenue for future research would be to evaluate all of the factors involved in children’s moral development.’

“be fully generalizable, on the other hand,” Run-on sentence.

“Which mak

---

## [Decision Letter · Decision Letter 1]

11 Aug 2025

Dear Dr. Garduño-Espinosa,

Thank you for submitting your manuscript to PLOS ONE. After careful consideration, we feel that it has merit but does not fully meet PLOS ONE’s publication criteria as it currently stands. Therefore, we invite you to submit a revised version of the manuscript that addresses the points raised during the review process.

The reviewers have noted the corrections made to the manuscript based on previous reviews, but still feel that there are a number of issues pending before it can be accepted for publication. The authors should properly address all comments made by the reviewers.

We look forward to receiving your revised manuscript.

Kind regards,

Shrisha Rao, Ph.D.

Academic Editor

PLOS ONE

Journal Requirements:

Additional Editor Comments:

See detailed remarks by reviewers.

Reviewers' comments:

Reviewer's Responses to Questions

**Comments to the Author**

Reviewer #1: (No Response)

Reviewer #2: (No Response)

2. Is the manuscript technically sound, and do the data support the conclusions?

Reviewer #1: Partly

Reviewer #2: Partly

3. Has the statistical analysis been performed appropriately and rigorously?

Reviewer #1: Yes

Reviewer #2: I Don't Know

4. Have the authors made all data underlying the findings in their manuscript fully available?

Reviewer #1: Yes

Reviewer #2: Yes

5. Is the manuscript presented in an intelligible fashion and written in standard English?

Reviewer #1: No

Reviewer #2: Yes

Reviewer #1: Thank you for revising the manuscript, but it still has very confusing parts that need to be revised with more accurate attention.

First of all, the introductory part, even though it was expanded, but the reader is could be confused, as the authors did not highlight the main parts: topicality, research aim and the research problem. The reader is left by himself/herself to figure that out. The authors should clearly state these issues so that everyone has the same understanding.

Second, the methodological part is not clarified enough – it lacks theoretical substantiation and citations of scientific literature. This part is very complicated if we need to duplicate. There should be clearly stated and described (with scientific literature substantiation) these missing issues:

• Research data collection methods;

• Research data analysis methods;

• Sample selection criteria and methods;

• Etc.

I strongly recommend the authors look through other high quality published articles and see how they are constructed. Also, authors should use clear and unified terms to make the idea and text as clear as possible (e.g., “The first control case…”, but “The second control scenario“ lines – 47-53). Is “the case” and “the scenario” the same? It raises misunderstandings.

Third, the discussion part still seems weak. The authors should clearly state: implications for theory, implications for practice, limitations and further research. The authors should come back and check for these issues, highlighting them. Also it is essential to expand on limitations (it is important to use critical thinking) and on further research, which is way beyond qualitative research. As soon as the authors accurately highlight implications for theory and practice, the limitation section will be clear to them, and the accurately written limitations part will help to highlight further research. These issues are interconnected.

In general, all-important issues should be highlighted in the text so that readers can easily find them. It is recommended for the authors to stand in the readers shoes and imagine if they were the readers, reading the study for the first time, would it be easy to duplicate it. To this point, the manuscript must be revised once more and cannot be published as it is.

Reviewer #2: Many of the responses given are acceptable. However, a number of further points have arisen. The most serious, it seems to me, is the potential contamination of the data by the children being able to see (and hear) their mothers. I am not satisfied that this has been appropriately reported in the manuscript. Please see my further comments on it below and other issues that have arisen.

3: “Moral dilemmas arise when two or more of these principles come into irresolvable conflict.” Are moral dilemmas irresolvable? Why do you need this claim? I suggest removing the word ‘irresolvable’.

4: It is uncertain what if anything has changed in the red paragraph from the first version.

The authors mention a paper by ‘Green[e] et al.’, but there is no reference for it.

“through moral reasoning about costs and benefits, a utilitarian judgment can be reached, while judgments based on norms (deontological) would be through automatic emotional reactions (moral intuitions) (19).”

“automatic emotional reactions (moral intuitions)” This is not generally what ‘moral intuition’ means the ethical literature. Further, the authors are misrepresenting what is said by Greene (19). What they actually say is: “Finally, there is the view for which I will argue, that deontology is more emotionally driven while consequentialism is more “cognitive.” I hasten to add, however, that I don’t believe that either approach is strictly emotional or “cognitive” (or even that there is a sharp distinction between “cognition” and emotion). More specifically, I am sympathetic to Hume’s claim that all moral judgment (including consequentialist judgment) must have some emotional component (Hume, 1978)” [p. 41]

So, please revise your statement carefully to reflect accurately the nuance of Greene's position.

9: “Mothers and children were in separate areas within the waiting room, so they could neither hear nor see each other.” In your response to my question the authors say: “We tried to get the child to focus on the interviewers so that he wouldn't hear or see his mother. We consider that the answers could be altered if the child saw his mother.”

These are two distinct descriptions of the experimental situation. In the manuscript the authors say that it is not _possible_: ‘could neither hear nor see each other’. However, it clearly was possible because the child would see (or hear) their mother if the interviewer did not manage to keep their focus, as you say in your response to my question.

It is important that you provide the reader with an accurate picture of the exact experimental conditions. Was any measure taken of how much each child focused or did not focus? If not, then how do the authors know that the answers were not influenced, or the data not contaminated? This is potentially quite problematic for the study. Please consider this carefully, and provide your reader with the full and accurate information and an assessment of the effect that it did or could have had on the results.

**Do you want your identity to be public for this peer review?** For information about this choice, including consent withdrawal, please see our Privacy Policy

Reviewer #1: **Yes:** dr. Inga Iždonaitė-Medžiūnienė

Reviewer #2: No

---

## [Author Response · Author response to Decision Letter 2]

15 Oct 2025

Reviewer 1

Thank you for revising the manuscript, but it still has very confusing parts that need to be revised with more accurate attention. First of all, the introductory part, even though it was expanded, but the reader is could be confused, as the authors did not highlight the main parts: topicality, research aim and the research problem. The reader is left by himself/herself to figure that out. The authors should clearly state these issues so that everyone has the same understanding.

Answer: The requested changes have been made to make the Introductory part more straightforward.

Second, the methodological part is not clarified enough – it lacks theoretical substantiation and citations of scientific literature. This part is very complicated if we need to duplicate. There should be clearly stated and described (with scientific literature substantiation) these missing issues: • Research data collection methods; • Research data analysis methods; • Sample selection criteria and methods; • Etc. I strongly recommend the authors look through other high quality published articles and see how they are constructed.

Answer: Thanks for the comment. The Methodology section was restructured, aiming to make it more understandable and detailed. The "Sample selection criteria and methods" were detailed in the "participants" section on page 6; the sample size calculation was also included, along with references (page 6). The "Research data collection methods" is detailed in the "Application procedure" section (page 9), and the "Research data analysis" is detailed in the "statistical analysis" section (page 11), and references were added in this part as well.

Also, authors should use clear and unified terms to make the idea and text as clear as possible (e.g., “The first control case…”, but “The second control scenario“ lines – 47- 53). Is “the case” and “the scenario” the same? It raises misunderstandings.

Answer: Indeed, "the case" and "the scenario" refer to the same in the manuscript; to avoid confusion, we homogenize the terms throughout the manuscript.

Third, the discussion part still seems weak. The authors should clearly state: implications for theory, implications for practice, limitations and further research. The authors should come back and check for these issues, highlighting them. Also it is essential to expand on limitations (it is important to use critical thinking) and on further research, which is way beyond qualitative research. As soon as the authors accurately highlight implications for theory and practice, the limitation section will be clear to them, and the accurately written limitations part will help to highlight further research. These issues are interconnected.

Answer: According to the commentary, it began with a general discussion, then addressed the theoretical and practical implications (page 19), and then outlined the limitations and future research (page 22). We consider that with the new information, the discussion will be more nourished and clear and will give rise to new research.

In general, all-important issues should be highlighted in the text so that readers can easily find them. It is recommended for the authors to stand in the readers shoes and imagine if they were the readers, reading the study for the first time, would it be easy to duplicate it. To this point, the manuscript must be revised once more and cannot be published as it is.

Answer: Thank you for the comment, it's very accurate. We have highlighted all the essential parts, hoping that the text in its entirety will be more complete and understandable.

Reviewer 2

Many of the responses given are acceptable. However, a number of further points have arisen. The most serious, it seems to me, is the potential contamination of the data by the children being able to see (and hear) their mothers. I am not satisfied that this has been appropriately reported in the manuscript. Please see my further comments on it below and other issues that have arisen:

3: “Moral dilemmas arise when two or more of these principles come into irresolvable conflict.” Are moral dilemmas irresolvable? Why do you need this claim? I suggest removing the word ‘irresolvable’.

Answer: Okay, thanks for the observation. The change has been made (p. 3).

4: It is uncertain what if anything has changed in the red paragraph from the first version. The authors mention a paper by ‘Green[e] et al.’, but there is no reference for it. “through moral reasoning about costs and benefits, a utilitarian judgment can be reached, while judgments based on norms (deontological) would be through automatic emotional reactions (moral intuitions) (19).” “automatic emotional reactions (moral intuitions)” This is not generally what ‘moral intuition’ means the ethical literature. Further, the authors are misrepresenting what is said by Greene (19). What they actually say is: “Finally, there is the view for which I will argue, that deontology is more emotionally driven while consequentialism is more “cognitive.” I hasten to add, however, that I don’t believe that either approach is strictly emotional or “cognitive” (or even that there is a sharp distinction between “cognition” and emotion). More specifically, I am sympathetic to Hume’s claim that all moral judgment (including consequentialist judgment) must have some emotional component (Hume, 1978)” [p. 41] So, please revise your statement carefully to reflect accurately the nuance of Greene's position.

Answer: Thank you for the comment. The literature was reviewed again, and the paragraph was restructured to avoid confusion about what Green said in his theory. The quote is included (page 4). We agree with your view that cognitive and emotional processes cannot be separated and that both can be found in both utilitarian and deontological judgments, so we also include some of Greene and Haidt's theory within the discussion, in the section on theoretical implications (p. 19).

9: “Mothers and children were in separate areas within the waiting room, so they could neither hear nor see each other.” In your response to my question the authors say: “We tried to get the child to focus on the interviewers so that he wouldn't hear or see his mother. We consider that the answers could be altered if the child saw his mother.”These are two distinct descriptions of the experimental situation. In the manuscript the authors say that it is not _possible_: ‘could neither hear nor see each other’. However, it clearly was possible because the child would see (or hear) their mother if the interviewer did not manage to keep their focus, as you say in your response to my question.It is important that you provide the reader with an accurate picture of the exact experimental conditions. Was any measure taken of how much each child focused or did not focus? If not, then how do the authors know that the answers were not influenced, or the data not contaminated? This is potentially quite problematic for the study. Please consider this carefully, and provide your reader with the full and accurate information and an assessment of the effect that it did or could have had on the results.

Answer: An apology for the confusion; our previous answer was not properly structured. In effect, children were separated from their mothers when asked to address ethical dilemmas. The comment about keeping their attention on the interviewer referred to them not worrying about not seeing or hearing their mother and wanting to look for her. Of course, before separating them, they were told what would be done and where their mother would be while they met with the interviewers. It is briefly explained in the "Application Procedure" section on page 10.

---

## [Decision Letter · Decision Letter 2]

9 Nov 2025

Early Ethics: Exploring moral intuition and maternal Influence in preschool children

PONE-D-25-08141R2

Dear Dr. Juan Garduño-Espinosa,

We’re pleased to inform you that your manuscript has been judged scientifically suitable for publication and will be formally accepted for publication once it meets all outstanding technical requirements.

Kind regards,

Luca Valera

Academic Editor

PLOS ONE

Additional Editor Comments (optional):

Please change your paper according to the referee's suggestions. Please send a new sanitized version of the manuscript.

Reviewers' comments:

Reviewer's Responses to Questions

**Comments to the Author**

Reviewer #1: (No Response)

Reviewer #2: (No Response)

2. Is the manuscript technically sound, and do the data support the conclusions?

Reviewer #1: Partly

Reviewer #2: Yes

3. Has the statistical analysis been performed appropriately and rigorously?

Reviewer #1: Yes

Reviewer #2: I Don't Know

4. Have the authors made all data underlying the findings in their manuscript fully available?

Reviewer #1: Yes

Reviewer #2: Yes

5. Is the manuscript presented in an intelligible fashion and written in standard English?

Reviewer #1: Yes

Reviewer #2: Yes

Reviewer #1: Thank you for addressing the comments, but the methodological part is still not clear enough. The "dyad" concept was not properly explained. In previous research a dyad is likely 2 participants, but the the authors research seems to be one. This is not good to have the misunderstanding, should be definitely clarified. Also to substantiate the research problem only one source is used. It can not be like that. Methods part needs more attention according to methodological principals that are general rules for all. Thank you

Reviewer #2: I am generally happy with the authors responses, but errors remain. Please check the manuscript thoroughly before publication.

Page 4: Again, to repeat, the authors are not referencing correctly. 1. The cited author’s name is ‘Greene’ not ‘Green’. You even make this mistake in your response. 2. The authors mention ‘Green et al’, but only a paper by Greene (i.e., not et al.) appears in the references list. Fix 1 and 2 throughout.

Line 93: What other authors? Provide references.

Line 94: “judgement utilitarian” is a bit odd in English, “utilitarian judgment”.

Page 22, Line 452: Missing full stop.

**Do you want your identity to be public for this peer review?** For information about this choice, including consent withdrawal, please see our Privacy Policy

Reviewer #1: **Yes:** dr. Inga Iždonaitė-Medžiūnienė

Reviewer #2: No

---

## [Editor Report · Acceptance letter]

PONE-D-25-08141R2

PLOS One

Dear Dr. Garduño-Espinosa,

I'm pleased to inform you that your manuscript has been deemed suitable for publication in PLOS One. Congratulations! Your manuscript is now being handed over to our production team.

Kind regards,

on behalf of

Dr. Luca Valera

Academic Editor

PLOS One